# Extrusion Simulation for the Design of Cereal and Legume Foods

**DOI:** 10.3390/foods11121780

**Published:** 2022-06-16

**Authors:** Magdalena Kristiawan, Guy Della Valle, Françoise Berzin

**Affiliations:** 1INRAE, UR 1268 Biopolymers Interactions and Assemblies (BIA), 44316 Nantes, France; guy.della-valle@inrae.fr; 2Université de Reims Champagne Ardenne, INRAE, FARE, UMR A 614, 51097 Reims, France; francoise.berzin@univ-reims.fr

**Keywords:** cellular structure, flow modeling, starch depolymerization, protein aggregation, shear viscosity, product design

## Abstract

A 1D global twin-screw extrusion model, implemented in numerical software, Ludovic^®^, was applied to predict extrusion variables and, therefore, to design various starchy products with targeted structure and properties. An experimental database was built with seven starchy food formulations for manufacturing dense and expanded foods made from starches, starch blends, breakfast cereals, pulse crop ingredients such as pea flour, fava bean flour, and fava bean starch concentrated, and wheat flour enriched with wheat bran. This database includes the thermal and physical properties of the formulations at solid and molten states, melt viscosity model, extruder configurations and operating parameters, and extruded foods properties. Using extrusion and viscosity models, melt temperature (T) and specific mechanical energy (SME) were satisfactorily predicted. A sensitivity analysis of variables at die exit was performed on formulation, extruder configuration, and operating parameters, generating the extruder operating charts. Results allowed the establishment of relationships between predicted variables (T, SME, melt viscosity) and product features such as starch and protein structural change, density and cellular structure, and functional properties. The extrusion operating conditions leading to targeted food properties can be assessed from these relationships and also the relationship between extrusion operating parameters and variables provided by simulation.

## 1. Introduction

Many starch-based foods, such as texturized ingredients, breakfast cereals, snacks, crackers, pasta, noodles, pet food, and many others, are produced by extrusion processes [1]. Starches and their blends are extruded to gain desirable end-usage properties. Ready-to-eat breakfast cereals are still the market leader in extruded foods. Society is leaning increasingly towards more sustainable and healthy diets by transitioning from animal to plant protein sources and enriching formulations with legume proteins and dietary fibers. Despite their nutritional advantages (high protein content, low glycaemic index) compared to cereal-based extruded foods, legume-based extruded foods are still rare. The growing demand for extruded snacks (4.4% per year in the market) makes the development of healthy extruded snacks even more desirable [2].

Under mechanical stresses and heat developed between screw and barrel during the extrusion process, the biopolymers undergo structural modifications: disruption of starch granules, melting of starch crystals, depolymerization of starch macromolecules, denaturation and aggregation of proteins, depolymerization of dietary fibers, and product browning due to Maillard reactions [3,4,5]. These changes are very important because they provide extruded starchy foods with an attractive appearance, structure, and relevant functional properties.

Due to an abrupt pressure drop at the die exit, the flash evaporation of superheated moisture forms a porous and expanded melt structure that is set after crossing its glass transition temperature. The structural features of the expanded foods govern their texture, functional and nutritional properties, and consumer acceptability [6,7,8,9,10,11,12]. The structural features consist of density, cellular structure, and properties of constitutive material, i.e., cell walls [13]. A phenomenological model of expansion allowed us to link extruded food features, such as expansion indices and cellular fineness to melt shear viscosity [14,15]. The melt viscosity takes into account the effect of formulation, such as fiber, protein, and starch contents, amylose to amylopectin ratio, and biopolymer structural changes during processing [16].

The development of mechanistic models of extrusion, based on the continuum mechanics approach, provides many benefits: a better understanding of the extrusion process and the development of computer-aided tools in simulation, optimization, and scale-up. In twin-screw extrusion modeling, two main approaches can be considered. In the first approach, local flow can be described with accuracy in a limited portion of the extruder by sophisticated numerical 2D or 3D models based on the finite element method (FEM) [17,18,19,20,21,22]. It can provide a very accurate description of the flow field only in a specific part of the extruder, and it is expensive in terms of computational resources and time.

The second approach is a global one; it is based on a simplified flow assumption and provides average values of the local flow variables (such as temperature, total dissipated energy, pressure, viscosity, filling ratio, and mean residence time) of the whole process, along the screws, from the hopper to the die exit [23,24]. It is based on various assumptions: (1) the flow is locally 1D or 2D; (2) the melting occurs instantaneously just before the first restrictive screw element; (3) the molten material behaves locally as a Newtonian fluid [24]. Computation of the flow parameters is done separately for each type of screw element and for the die components. The elementary models are linked together to obtain a global description of the flow along the extruder. This 1D global model has been implemented into a software called Ludovic^®^ v7.0.0 Classic Edition (Sciences Computers Consultants, Saint Etienne, France). It is quite straightforward to provide predictions of flow variables with simple computing resources and time: about a few seconds per simulation using a laptop configured with Intel^®^ Core™ i7-7600U processor (dual core). A fairly good agreement was obtained between simulation results from this 1D global model and a full 3D FEM model [18].

Despite significant progress in modeling and simulation, the design of extruded products at the industrial level is still based on a trial-and-error approach. One of the main challenges is determining the viscous behavior of melts under extrusion-like conditions that require specific rheometers, e.g., an in-line slit die rheometer or pre-shearing capillary rheometer (Rheoplast^®^) (Société Courbon, Saint-Etienne, France) [25,26]. This difficulty can be circumvented, in the first approach, by assuming that starch viscosity is a function of amylose content, and it can be interpolated from the published data on maize starches having several amylose contents [27].

The 1D global model has been widely used for the design of experiments of processing, but it is still limited to simple formulation (maize and wheat starches and wheat flour) [23,28,29,30]. In process design, the model is exploited as a numerical screening tool of operating conditions to define and optimize the experimental domain in terms of flow variables (melt temperature, specific mechanical energy (SME), pressure). Its implementation in product design, for a wide range of complex formulations encompassing various structural and functional properties for one product, has never been tackled before. Until now, a simple link between a single property of extruded starchy products (for example, hydrosolubility) and a single process variable (for example, SME) was considered [29]. In this context, the aim of this paper is to test whether the 1D global extrusion model, implemented with an appropriate viscosity model, can be deployed as a prototype of a computer-aided tool for predicting properties of extruded foods and structural changes of biopolymers based on realistic formulations, from operating conditions. In other words, it will serve for product design applications.

For this purpose, we selected seven starchy food formulations to manufacture dense and expanded foods: cereal and tuberous-based pure starches and their blends with a wide range of amylose–amylopectin ratio, legumes-based ingredients (pea flour, fava bean flour, and fava bean starch concentrate), breakfast cereals, and blends of wheat flour–wheat bran. A detailed experimental database was built, including input and output data for extrusion simulation and extruded product structural and functional features. After describing the 1D extrusion model, virtual extrusion trials were performed, and validation steps were systematically detailed. The extruder operating curves representing a variety of predicted extrusion variables in the function of operating parameters were built. Then we strived to show how products with targeted structural and functional properties can be manufactured using the relations between extruder operating charts, predicted extrusion variables, and extruded product features, including biopolymer structural changes.

## 2. Materials and Methods

### 2.1. Building Database

Database for twin-screw extrusion simulation has been collected from literature and our own experimental results [31,32,33,34,35,36]. Due to the large number of experimental works, case studies were selected by applying several criteria: (1) presence of melt viscosity model and melting transition data as input, (2) existence of experimental data of melt temperature and specific mechanical energy (SME), and (3) properties of final products exhibiting clear relations with extrusion variables.

#### 2.1.1. Raw Materials

Seven food recipes have been selected, reflecting a wide range of complexity of food formulations. The recipes were extruded to manufacture foods from the simplest ones, e.g., dense potato starch [32] and expanded starches (high amylose maize, waxy maize, and their blends revealing a broad variation of amylose content [27,33], and potato [31], to complex ones, e.g., breakfast cereals, and expanded snacks made from (1) legumes-based ingredients (pea flour [34], fava bean flour, and fava bean starch concentrate), and (2) blends of wheat flour and wheat bran [35,36,37] (Table 1). The extrusion data set of fava bean ingredients and breakfast cereals were produced from our original works. For these cases, raw material chemical composition was determined according to standard methods [34].

Two breakfast cereal foods were obtained by extrusion of a cacao-based formulation (called CB/Choco Ball) composed of a blend of rice flour, wheat flour, cacao powder, sugar, and salt and of a honey-based formulation (HB/Honey Ball) composed of a blend of corn flour, malt extract, sugar, and salt.

An overview of the chemical composition of all extrusion feeds is given in Table 1.

#### 2.1.2. Extrusion Trials

The extrusion variables cover a broad spectrum of moisture content (15–36% wet basis), melt temperature (85–190 °C), and specific mechanical energy (100–5500 kJ/kg). An overview of extrusion systems for all feeds is given in Table 2, including equipment specificities.

Maize and potato-based expanded starches (Maize A, Maize B, Maize C, Maize D, and Potato 1; case n°1) and wheat-based snacks (WF, WF-LB, WF-HB; case n°7) were manufactured using pre-pilot extruders having a screw diameter between 25 and 55.5 mm [27,31,33,35,36]. The screw profile included conveying screw elements followed by a left-handed screw element. While maize starch blends were extruded through a long-slit die, the extrusion of potato starch and wheat flour-wheat bran used a short circular die. Pea flour extrusion (PF; case n°5) [34] used a more complex screw profile: conveying elements followed by three kneading discs staggered at 45°, and then three left-handed elements. Several die geometries were used, circular (D = 3 mm) of various lengths (5, 10, and 15 mm), attached or not with a multi-step slit die rheometer, as described in detail in Appendix A (Figure A1).

Dense pea and potato starch (PS, Potato 2; case n°3) [32] and fava bean-based snacks (FF, FSC; case n°6) were extruded using laboratory-scale extruders with a screw diameter of 11 and 16 mm and equipped with a cylindrical die. The screw profile included conveying elements, followed by kneading discs staggered at 90° and one left-handed screw element. For manufacturing dense (non-expanded) products (PS, Potato 2), the temperatures of the last barrel (Tb) and die (Td) were regulated at 80–95 °C, otherwise, the temperatures were set at melting temperature level.

Extrusion of breakfast cereal formulations (HB and CB; case n°4) was carried out on a pre-pilot co-rotating twin-screw extruder (Coperion Werner & Pfleiderer ZSK 26Mc) equipped with a circular die. The same extruder and screw configuration, and temperature profile of barrel and die were used, as for pea snacks manufacturing [34]. The die had a diameter of 3 mm and a length of 15 mm.

### 2.2. Simulation of Extrusion Trials Using Ludovic^®^ Software

#### 2.2.1. Extrusion Modelling

The extrusion simulation is based on a global model developed by Vergnes et al. [24]. This model uses a 1-D approach under the assumption that molten material behaves locally as a Newtonian fluid.

The flow is described by the Stokes equations in cylindrical coordinates (r, θ, z). Each conveying element is considered as a C-chamber associated with an angular sector representing the intermeshing area. Assuming that temperature and viscosity are locally constant, the velocity field can be determined in each element, and the volumetric flow rate $Q_{v}$ through the screw channel is expressed as:(1)$$Q_{v}=A\Omega +B\frac{1}{\eta }\frac{\Delta P}{\Delta \theta }$$
where A and B are coefficients depending on screw geometry, Ω is the screw rotation speed (rad·s^−1^), ΔP is the pressure variation (in Pa) along angular screw section Δθ. The first term on the right side stands for drag flow, and the second one for pressure-driven flow. The non-Newtonian behavior of the molten material is taken into account by applying an appropriate model of shear viscosity $\eta \left(\dot{\gamma }\right)$ into Equation (1), where $\dot{\gamma }$ is the local shear rate. Specific models have also been developed for reverse screw elements [38] and blocks of kneading discs [39].

Flow equations are solved for each section of the screw and the die. These solutions are linked together to obtain a global description of the molten material flow along the extruder. The temperature changes ΔT in a section are obtained from the energy balance of accumulation, viscous dissipation $E_{v}$, and heat transfer by conduction to the barrel $E_{\mathrm{cd}}$:(2)$$\mathrm{Cp}\;\Delta T+E_{\mathrm{cd}}=E_{v}=\frac{\dot{W}}{\rho_{m}Q_{v}}$$
(3)$$\mathrm{with}\;\dot{W}=\int_{V}\eta {\dot{\gamma }}^{2}\mathrm{dV}$$
where $\dot{W}$ is the power dissipated in the volume V of a screw element, ρ is the material melt density, and $C_{p}$ is its specific heat.

The melting zone is determined according to experimental observations: it is located at the first restrictive element (left-handed element, kneading disc), and melting is assumed to be instantaneous [40]. It is supposed that, at the point where the pressure starts to increase, upstream of the first restrictive element, the material is in the molten state and its temperature is equal to the melting temperature (T_m_).

The computation starts from the die exit and proceeds backwards. As the final product temperature is unknown, an iterative procedure is used. A final temperature of the product at the die exit is chosen, and the pressure drop inside the die is calculated to go back to the upstream pressure value (to the upstream screw elements). The evolutions of pressure and temperature are thus determined element by element (section by section), from downstream to upstream until the melting point. The temperature calculated at this point is then compared with the melting temperature: if they are equal, the calculation is completed; otherwise, the final temperature is modified, and the calculation is repeated until convergence.

#### 2.2.2. Input Variables

The input data are divided into three groups. The first group refers to extruder configuration. It covers screws, barrels, die, and also the position of specific zones (feeding, degassing). Examples of the screw profiles, for all feeds composed of right-handed and left-handed screw elements, and blocks of kneading discs are given in Appendix A (Figure A2). Dies are characterized by elementary elements (e.g., slit, pipe, associated in series or parallel).

The second group refers to food properties:(1)Physical and thermal properties

They cover density ρ, heat capacity $C_{p}$, and thermal conductivity k in the solid and molten state, and melting temperature and enthalpy. Their dependence on moisture content (wet basis) can be calculated according to additive rules from the values for water and dry molten starches (Appendix A, Table A1) [41,42,43]. Most results on the melting temperature (T_m_) as a function of moisture content of studied materials can be retrieved from the literature [32,34,42,44,45,46,47] (Appendix A, Figure A3). The melting point of maize starches, having different amylose content (5%, 23%, 47%, and 70%), were obtained from the data set of waxy maize (amylose content of 5%); wheat (25%), smooth pea (35%), and wrinkled pea (75%) starches. The melting temperature dataset of breakfast cereal formulations (HB, CB), and fava bean flour and starch concentrate were determined experimentally as the peak of the DSC melting endotherm using standard procedure [32,48] (Appendix A, Figure A3).

(2)Rheological properties of molten starchy foods

The variation of starchy melt viscosity (η) with shear rate $\left(\dot{\gamma }\right)$ is generally described by the Ostwald–de Waele power law [25]:(4)$$\eta =K{\dot{\gamma }}^{n-1}$$
where n is the flow index (0 ≤ n ≤ 1) and K (Pa·s^n^) the consistency.

The parameters K and n were expressed as a function of temperature (T, in °C), moisture content (MC), and SME according to [27]:(5)$$K=K_{0}\exp\left(\frac{E}{R}\left(\frac{1}{T_{a}}-\frac{1}{T_{0}}\right)-\alpha \left(\mathrm{MC}-{\mathrm{MC}}_{0}\right)-\beta \left(\mathrm{SME}-{\mathrm{SME}}_{0}\right)\right)$$
(6)$$n=n_{0}+\alpha_{1}\;T+\alpha_{2}\;\mathrm{MC}+\alpha_{3}\;\mathrm{SME}+\alpha_{4}\;T\;\mathrm{MC}+\alpha_{5}\;T\;\mathrm{SME}+\alpha_{6}\;\mathrm{MC}\;\mathrm{SME}$$ where $K_{0}$ is the consistency at reference conditions, E is the activation energy, R is the gas constant, $T_{a}$ is the absolute temperature (in K), α is the water plasticization coefficient, β reflects the thermomechanical history coefficient, and $n_{0}$ and $\alpha_{i}$ are constants accounting for the dependences of the flow index on T, MC, and SME and their interactions. The other constants, ${\mathrm{MC}}_{0}$, $T_{0}$, and ${\mathrm{SME}}_{0}$, stand for reference moisture content (10%), temperature (353 K), and SME (350 kJ/kg).

The viscosity model parameters of starchy melts, e.g., maize starches having different amylose content, potato starch, breakfast cereals, and blend of wheat flour–wheat bran were retrieved from the literature and are presented in Table 3. Under the hypothesis that viscous behavior is governed only by the amylose content of the starch, the viscosity of fava bean starch concentrate was approached by maize starch having an amylose content of 47%. The viscosity behavior of pea starch and fava bean flour was approached by the pea flour. Indeed, a previous study revealed that the master curve of shear viscosity of these feeds, determined by a pre-shearing capillary rheometer (Rheoplast^®^) (Société Courbon, Saint-Etienne, France), were very close to each other [49].

Finally, the third group of data concerns the operating conditions of the extruder. This data covers the barrel and die temperature profile, the moisture content, the total flow rate, the screw speed, and the heat transfer coefficients between molten material and barrel, screws and die, expressed through a Nusselt number. The data was overviewed in Table 2 for all recipes. For all simulations, regardless of the recipes, the Nusselt number was set to 20.

#### 2.2.3. Output Variables

The main variables predicted by the 1D global extrusion model (Ludovic^®^ v7.0.0 Classic Edition) (Sciences Computers Consultants, Saint Etienne, France) are melt temperature T, SME, pressure, and viscosity. These variables can be linked to the biopolymers transformation (e.g., starch melting and depolymerization, and protein aggregation), structural (e.g., expansion) and functional properties (e.g., water solubility index of starch) of extruded foods, considered as indirect output variables. Starch structural changes were represented by some features, such as the number of remnant starch granules/unit area, residual molar mass of starch, and intrinsic viscosity. Protein cross-linking by covalent bonds other than disulphide bonds was quantified by the amount of in-extractable proteins in sodium dodecyl sulphate (SDS) and dithioerythritol (DTE). Expansion behavior was characterized by density, sectional expansion index, and cell density (number of cells per volume unit). These properties have been listed in Table 4, hence completing the database, with their origin (literature, this work) and measurement methods. Actually, when these features were not available in the literature, they were characterized especially in this present work. In this case, methods have been detailed in Section A.2.

## 3. Results and Discussion

### 3.1. Validation of Extrusion Simulation

The reliability of the extrusion simulation was confirmed: (1) by comparing computed extrusion variables with the measured ones (T and SME) (Figure 1), (2) by studying the realistic feature of the axial profile of computed flow variables along the screw (Figure 2 and Figure 3), and (3) by examining in detail the responses of computed flow variables, upon the change of operating conditions (Figure 3 and Figure 4).

#### 3.1.1. Comparison between Experimental and Simulation Results

Like in the experimental trials, the predicted melt temperature T_com was retrieved at the die exit. For most simulations, the computed temperature is overestimated by 5–10% compared to the measured temperature. The uncertainty of measurement can explain this difference. Temperature sensors are flush-mounted on the die wall with little penetration into the melt; consequently, the measured temperature differs from the actual one at the core of the product. Nevertheless, the computed temperature correlates well with the measured temperature, regardless of the formulation and extrusion system (*R^2^* = 0.88) (Figure 1a). Good correlations were also obtained between predicted and measured SME values (*R^2^* = 0.71–0.96), depending on the formulation (Figure 1b). The predicted SME is underestimated by 35–65% because the extrusion model only considers the melting energy and the viscous dissipation energy after the melting section, and neglects the solid transport and solid friction energy (inter-particle and particle/metal, metal/metal), which depend on raw material. All these trends in predictions: underestimation of SME and overestimation of temperature are in agreement with published results on extrusion simulation of starches and wheat flour [28,30]. The fair predictions of T and SME confirm the pertinence of the extrusion model for the design of extruded starchy products.

#### 3.1.2. Axial Profile of Predicted Flow Variables

Simulation results were illustrated for two recipes of breakfast cereals taken as examples by presenting the axial profiles of the predicted variables (melt temperature, specific mechanical energy (SME_com), pressure, and viscosity) along the screws (Figure 2a) for typical moisture, feed rate and screw speed conditions (Figure 2 and Figure 3). These profiles give us insight into thermomechanical conditions along the screws. The results obtained for other recipes exhibit similar trends and are not presented here.

The temperature increased progressively to values superior to the melting temperature T_m_ (T_m_ = 133 °C for HB and T_m_ = 165 °C for CB) along the kneading discs and rose sharply along the left-handed elements (Figure 2b). The melt flow across the kneading discs caused a temperature increase of about 5 to 10 °C, depending on the material and flow conditions. A more severe temperature step was observed for left-handed elements: 35 to 60 °C for the three elements. The increase in temperature was essentially due to intense shear stresses across the restrictive elements causing high viscous dissipation. The temperature reached a maximum at the outlet of the last left-handed element. Then it decreased slightly in the melt conveying section before a final slight increase through the die.

The cumulative SME_com was calculated by summing values in each screw element. The melt flow across the restrictive elements (kneading discs and left-handed screw elements) increased the SME_com (Figure 2c), particularly in the left-handed elements. The sudden increase in SME_com was concomitant with the step increase in temperature, underlining the importance of viscous dissipation, which was mainly governed by the restrictive screw elements.

The axial pressure profile shows that the main part of the extruder was only partially filled since the relative pressure was zero for screw length between 50 and 200 mm (measured from the die) and beyond 300 mm. The left-handed screw elements and the right-handed screw element in front of the die were the only zones where the melt was under pressure (Figure 2d).

Since the viscosity depends on temperature and SME, the melt viscosity profile along the screws is also related to temperature and SME profiles (Figure 2b,c,e). At the melting section (in the first kneading bloc), the product temperature and SME were relatively low, and consequently, the local viscosity was high. In the following restrictive elements, and the conveying section nearby die inlet, there was an increase in temperature and SME due to the increase in viscous dissipation, leading to a decrease in viscosity. The knowledge of melt viscosity, mainly at the die exit, is very important, because it can be hardly measured experimentally and it partly controls the melt expansion at die exit [14].

The singularities of variable profiles (pressure peak, changes of temperature slope and energy increase and viscosity decrease) meet exactly the position of restrictive elements. This fact underlines the importance of viscous dissipation, mainly governed by the location of restrictive screw elements. Similar trends were observed for extrusion simulation of starches [28,29]. The CB recipe, once fully molten, exhibits higher temperature, SME, viscosity, and pressure along the whole screw length. Brugger et al. [50] observed that at close T and MC, the CB exhibited higher viscosity than the HB.

According to Figure 2b, the maximum temperature is predicted at the last restrictive element, i.e., at the position where it is difficult to measure it. For CB, at the selected operating condition (MC = 15%, N = 500 rpm, Q = 20 kg/h), the maximum temperature was high (230 °C) which suggests that food degradation may occur. For HB, under the same conditions, the maximum temperature was lower, but still very high for food stuffs (185 °C).

The importance of viscous dissipation on thermo-mechanical history was also proven by the negative effect of moisture content on the axial profile of all major flow variables (Figure 3a–c). When the moisture content is increased from 19% to 23%, the magnitude of temperature and SME_com is reduced all along the screws by a factor of 1.5 and 3, and pressure underwent a larger drop at the die entrance, by a factor of 4.5. The viscosity dependence on moisture content can explain this. Higher moisture content leads to a lower melt viscosity, which decreases energy dissipation and then lowers temperature and pressure.

Another important variable for a twin-screw extruder is the filling ratio, which is proportional to Q/N (Figure 3d). For each moisture content, the predicted SME_com of the CB recipe is negatively correlated with Q/N, following a power function. The same trend describing the effects of Q/N on SME was observed for experimental results of twin-screw extrusion of starchy products [28,32,34]. In fact, SME is usually proportional to N/Q: an increase of screw speed (N) or a decrease in feed rate (Q) will induce torque reduction, due to the shear-thinning behavior of melt, as well as the lower filling ratio of the extruder and more important viscous heating. According to Figure 3d, moisture content also affects SME-filling ratio curves negatively, which is explained by the influence of the water content on the viscosity.

#### 3.1.3. Sensitivity Analysis

The impact of changing one operating conditions of the three ones (screw speed N, total flow rate Q, last barrel temperature Tb) on extrusion variables was examined systematically, while keeping other conditions (i.e., extruder configuration, screw profile) constant. Results are presented as 2D-operating charts and illustrated in Figure 4a–d for the manufacturing of breakfast cereal (HB recipe) at a moisture content of 19%.

As indicated previously, SME_com increases with screw speed and decreases with flow rate (Figure 4a). For each Q, SME increases by 20–30% when N increases from 150 to 700 rpm. At any N, SME drops by 35–40% when Q increases from 10 to 50 kg/h, with the highest decline occurring at the beginning of Q change (from 10 to 20 kg/h). The variations of T at die exit (T_com), when N and Q are changed, follow the same trend as SME_com, but it is quite insensitive to the change of Q (Figure 4b). The inverse trend with N and Q was observed for computed die entrance pressure (P_com): it decreases with N and increases with Q by a factor of 1.75–2.5 (Figure 4c). The negative impact of Q was also observed on predicted melt viscosity at the die exit: η_com drops by a factor of 2–2.25 (Figure 4e). However, an increase of Tb from 90 to 140 °C leads to a decline in η_com by a factor of 1.2–1.35. The increase of screw speed and the decrease of flow rate positively affects the viscous dissipation and, therefore, the SME and melt temperature. Indeed, the melt temperature increase leads to a decrease in the viscosity, and consequently, pressure.

Finally, the effect of conductive heating was studied on melt temperature. Regardless of N, an increase in barrel temperature (Tb), from 90 to 140 °C, leads to melt temperature (T_com) rising, by 5–10%, following linear trends (Figure 4e). However, the melt temperature is always higher than that of the barrel. It proves that viscous dissipation has more impact than the heat transfer between barrel and melt.

The set of Figure 4b–d can be used to evaluate the upper limit of operating pressure and melt temperature. For example, let us fix the upper limit of maximum melt temperature and pressure to be 165 °C and 10 MPa. The feasible operating domain for barrel temperature (Tb) of 90 °C at the highest flow rate (Q = 50 kg/h) is then screw speed in the interval of 300–550 rpm.

The order of magnitude of predicted extrusion variables and the trend of their responses to changes of extruder operating conditions and screw elements are similar to those observed for the extrusion of standard cereal and starches [28,29,56]. The similarity in the trend of responses was found for all studied starch-based recipes, likely because the responses are mainly governed by the viscous (shear thinning) behavior of the molten starch. These facts validate the 1D global extrusion model for the design of starchy foods. In addition, the 1D global model of twin-screw extrusion allows us to better understand the machine (extruder) and extrusion process itself. The knowledge of temperature profile and the possibility of adjustment of processing conditions using simulation appears thus advantageous, particularly for thermo-sensitive foods.

### 3.2. Application in Food Design

In this section, extrusion modeling is applied to determine the operating conditions that lead to a product with desired structure and properties. This approach is based on: (1) relationships between computed extrusion variables and product features (Figure 5a, Figure 6a,b, Figure 7a,b and Figure 8a–c, and Appendix A, Table A2 for equations) and (2) using these relations and the global extrusion model to select an interval of operating conditions that permit to obtain the desired properties

#### 3.2.1. Intrinsic Viscosity of Expanded Starches

Starch transformation was assessed by the intrinsic viscosity, reflecting the extruded product’s molar mass. It was normalized to the property of native starch, resulting in a relative drop of intrinsic viscosity ΔVis (%). The higher ΔVis, the larger the extent of degradation, the more depolymerized extruded starch, and the lower the molar mass is. Extrusion conditions for expanded maize and potato starches (Maize A, Maize B, Maize C, Maize D, and Potato1) are indicated in Table 2.

For all recipes, an increase in computed SME (SME_com) led to a drop in intrinsic viscosity ΔVis between 20 to 80%, which can be interpreted by starch depolymerization (Figure 5a). Conversely, no clear trend was observed for the impact of melt temperature (not shown). The trends are in accordance with experimental results (Appendix A, Figure A4a). The effect of viscous dissipation favors the disruption of starch granules and macromolecular chain splitting has been observed many times before, e.g., by Li et al. [3]. The relationships were manifested fairly well by power trends (*R^2^* ≈ 0.7–0.85), the parameters of which depend on starch origin (Figure 5a). Their numerical values are given in Appendix A, Table A2. It can also be seen that, in the SME_com interval [250, 500 kJ/kg], the values of ΔVis rank inversely to their amylose content for maize starches. This is due to the larger molar mass of amylopectin, a highly short-branched macromolecule, which makes it more sensitive to shear, therefore to SME, than amylose, a linear macromolecule (10^8^ vs. 2 × 10^6^ g/mol). The extent of degradation ΔVis is the most important in the case of potato starch, from 60% to 80% in the interval SME_com [250, 350 kJ/kg] (Figure 5a). This trend can be explained by the higher value of intrinsic viscosity for its native starch (450 mL/g) compared to the values given for normal maize starch (amylose content of 23%) in the native state (22.0 mL/g) [31]. Therefore, the larger size (molar mass) of potato starch macromolecules, amplified the sensitivity of this starch to shear.

Since it is possible to predict macromolecular degradation from a computed extrusion variable, it is interesting to predict the conditions for processing maize starch, e.g., having an amylose content of 47%, with a targeted intrinsic viscosity drop ΔVis = 50 ± 5%. According to Figure 5a, SME_com should be in the interval [625, 875 kJ/kg]. Besides operating conditions (Q, N), viscous dissipation, reflected by SME, is sensitive to the profile and geometry of screw elements, mainly restrictive elements. Therefore, simulations were conducted to study the effect of the length of the left-handed screw element: (a) 0 mm, (b) 25 mm, and (c) 50 mm. According to the response surfaces, or operating chart (Figure 5b), at feed MC = 20%, when the restrictive element was absent (0 mm, case a), even by using most severe conditions (highest N and lowest Q), SME_com was lower than 310 kJ/kg, hence, lower than the target. In case b, when its length is 25 mm, the extruder can only operate in a very restrictive interval of Q and N values, represented by the hatched area: N ≥ 650 rpm and Q < 11 kg/h, which leads to SME_com lower than 665 kJ/kg, hence in the target interval (Figure 5c). Finally, for the longest restrictive element (50 mm, case c), the operating window was larger, with feed rate Q possibly reaching 20 kg/h for larger N values (750 rpm) (Figure 5d). This example underlines how the simulation can be used to select the relevant screw profile, which is helpful from a practical viewpoint, e.g., before implementing experimental trials.

#### 3.2.2. Partially Melted Non-Expanded Starches: Study of Starch Structural Modification

Here, we deal with specific products, namely partially melted (partially gelatinized) dense extruded starches, i.e., Potato 2 and pea (PS). This kind of product was obtained through extrusion at the moderate moisture content (30%) and at the low barrel and die temperatures, i.e., 75–95 °C (Table 2, Appendix A, Figure A3). The starch transformation indicator was accessed at two levels: (1) granular one, expressed by the number of granular starch remnants per mm^2^ (Ng), and (2) molecular one, revealed by the residual molar mass (RM, %) that is obtained by normalization to native starch property.

For both starches, the Ng, reflecting the extent of starch granule breakdown or starch crystallite melting, diminished with an increase in SME_com following exponential functions (Figure 6a). This phenomenon is common for starches extrusion [3]. In contrast with experimental results (Appendix A, Figure A4b), the Ng prediction of extruded pea starch was poor (*R^2^* = 0.56) and the fitting parameters depended on the raw material; this is likely due to the underestimation of SME prediction of pea starch extrusion. This inaccurate prediction comes from approaching the viscous behavior of pea starch melt with pea flour as an input of the 1D global extrusion model. Indeed, a previous study revealed that the shear viscosity of these feeds, determined by a pre-shearing capillary rheometer (Rheoplast^®^) (Société Courbon, Saint-Etienne, France), operated at SME < 250 kJ/kg, was very close [49]. Prediction improvement could be achieved by determining the viscous behavior of pea starch on a larger SME interval. For potato starch, the Ng dropped by 95%, while SME_com increased by a factor of 4 (*R^2^* = 0.73).

When the SME_com increases, a drop in RM occurs concomitantly with a decrease in Ng (Figure 6a,b). It means that the starch macromolecular chain splitting phenomenon is also involved in the starch crystal melting process; both phenomena being intensified with the more severity of shear stress [32]. The trends, specific for each product, can be fitted fairly well with power functions (*R^2^* = 0.7–0.89). The general tendency is in accordance with experimental results; however it is obviously seen experimentally that potato starch is more sensitive to molecular degradation than pea starch (Appendix A, Figure A4c). The RM of potato product dropped by a factor of 4 when measured SME was increased by a factor 1.5. On the contrary, the RM of pea product diminished by a factor of only 1.5 when the SME_mes was intensified by a factor 4. This trend was not visible for the relationship between RM and computed SME due to underestimation of SME prediction. The higher sensitivity of potato starch macromolecules to shear can be explained by (1) its higher amylopectin content, which leads to a higher molar mass and lesser tendency to entangle and (2) higher extrusion temperature: [T_m_, T_m_ + 10 °C] vs. [T_m_−26, T_m_−16 °C].

One example of a commercial application of this kind of product is dried noodles. The functional properties of noodles, such as optimal cooking time, water uptake, and cooking loss, depend on the state of starch transformation. During cooking (boiling), the remnant starch granules of noodles uptake water and swell, and cooking loss occurs through the diffusion (leaching) of depolymerized amylose and amylopectin outward the ghost starch granules, from the core to the surface of noodles. Cooking loss can be minimized by allowing starch swelling without excessive breakdown/leaching of its macromolecules. In other words, a compromise between two properties, Ng and RM, shall be sought, based on the assumption that the higher the RM, the more enhanced the leaching.

As an example, we want to manufacture partially gelatinized potato starch with the following features: (1) highly amount of undisrupted starch granule: 150 < Ng < 300; (2) moderate level of RM (RM = 50 ± 10%). According to the relationship described in Figure 6a,b, extrusion should be conducted at SME_com = 525 ± 25 kJ/kg. The operating chart of the extruder is given in Figure 6c. The feasible domain of screw speed and flow rate conditions is presented as a hatched area. The targeted product can be obtained at a low flow rate (0.6 kg/h) by tuning the screw speed N at 40–50 rpm. When a high throughput is required (Q = 1.8 kg/h), the screw speed should be set at a higher velocity: 140–165 rpm.

#### 3.2.3. Legumes Based Snacks: Compromise between Protein Transformation and Expansion

Here, the features of protein-rich snacks, extruded from pea and fava bean ingredients, were represented by protein solubility, in terms of insoluble proteins and density. While the latter reveals the macrostructure, the first can reflect the structure at supramolecular level. The gain in insoluble proteins indicates a higher amount of proteins crosslinked by covalent bonds other than S-S ones, i.e., isopeptide bonds. The higher is the density and the less expanded is the snack.

The rising of melt temperature, at die exit, observed in the pea flour extrusion simulation promoted the protein cross-linking by non S-S bond (*R^2^* ≈ 80, Figure 7a), in accord with experimental results (Appendix A, Figure A4d). The onset of protein cross-linking occurred at a computed melt temperature (T_com) of 155 °C. The snack density decreased from 800 to 100 kg.m^3^, in other words, expansion was magnified by a factor of eight, when T_com increased from 125 to 175 °C, regardless of formulation (*R^2^* = 0.75, Figure 6b). The positive impact of temperature on legumes expansion, in agreement with experimental results (Appendix A, Figure A4e), is not common for extrusion of starches, where the inverse trend is usually observed [14]. The expansion mechanisms are likely governed by melt rheological changes linked to a certain morphology of starch-protein composite that is probably tuned by structural protein modification [34]. Numerical values of all fitting parameters are summarized in Appendix A, Table A2. The impact of SME_com on snack density and protein transformation is not clear (*R^2^* = 0.45 and 0.55, result not shown).

A low density (lower than 200 kg/m^3^), resulting from high-temperature extrusion (155 < T_com °C < 175 °C) at low MC (18% and 21%), is indispensable for obtaining crispy snacks. However, extrusion at these temperatures will be accompanied by a higher amount of insoluble proteins (from 5 to 25%). In addition, according to some authors, the protein cross-linking can deteriorate nutritional properties, such as, i.e., protein digestibility [10], and cause a decline in customer acceptance because protein cross-linking increases viscosity of chewed food [9], leading to more difficulty in food swallowing. Therefore, the design of protein-based snacks like pea snacks requires a compromise between density and protein cross-linking.

An example of compromise is set as follows: snacks having a density lower than 200 kg/m^3^ and insoluble proteins lower than 10% can be extruded at computed temperature (T_com): 155 < T_com °C < 165 (see correlation charts, Figure 7a,b). The operating charts of extruder for extrusion at MC of 21% are given in Figure 7c,d. The hatched area indicates possible combinations of Q and N, for Tb = 90 °C (Figure 7c). and sets of Tb and N, for Q = 20 kg/h (Figure 7d), to achieve targeted T_com. When Tb is fixed at 90 °C, the domain of N for a given Q are 600–650 rpm at the lowest Q (10 kg/h) and 625–700 rpm at the highest Q (50 kg/h). When Q is fixed at 20 kg/h, N can be tuned as: N = 625–700 rpm for low Tb (90 °C) and N = 365–500 rpm for high Tb (140 °C).

#### 3.2.4. Expansion of Wheat Snacks Enriched with Wheat Bran (Fiber)

The features of wheat-based snacks are represented by hydro-solubility of starch (WSI_starch_), macrostructure described by sectional (radial) expansion index (SEI), and cellular structure expressed as cell density per cm^3^ (NC). The higher the WSI_starch_, the more soluble are the starch molecules. This variable can thus indicate the state of starch depolymerization induced by extrusion. The higher the NC, the more porous the expanded structure.

Regardless of fiber (bran) content, all wheat flour-based snacks exhibited a unique linear and positive relationship between WSI of starch and SME_com (Figure 8a), in agreement with experimental results (Appendix A, Figure A4f). The higher the shear energy, the higher the extent of starch depolymerization, which is a common trend in starch extrusion [3]. More soluble and smaller starch molecules are more accessible to digestive enzymes during food digestion; however, a highly digested starch can lead to a high value of the glycaemic index.

The prediction of WSI_starch_ from both computed and measured SMEs is poor (Figure 8a and Figure A4f). If the fiber (wheat bran) is water-soluble, the data dispersion can be likely due to competition in water absorption between starch and fiber; otherwise, the dispersion origin can be explained by filler content (insoluble fiber). The increase of filler content in continuous molten starch can increase melt viscosity, affect viscous dissipation energy, and finally, the degree of starch transformation in the extruder [16]. The prediction could be improved by several means: First, presenting the WSI of total solids, considering fiber and starch, instead of the WSI of starch only. Second, determining the quantity of soluble starch experimentally using a more sophisticated method, such as orcinol titration [34] rather than the gravimetric-based method used by Robin et al. [36]. Third, improving the viscosity model, the effect of filler volume fraction could be taken into account.

The melt temperature and viscosity at the die exit was predicted by simulation since experimental data was not available. Acceptable correlations following exponential functions (*R^2^* = 0.7–0.8) were found between T_com and structure, such as SEI (indicator of expansion in the radial direction) (Figure 8b) and cell density (porosity indicator) (Figure 8c). However, the bran inclusion led to a similar trend but different correlations compared to the flour without bran. We observed that melt temperature and bran enrichment negatively affected SEI. In contrast to SEI, bran inclusion and melt temperature positively affected cell density (NC). For bran-enriched snacks, NC escalated by a factor of 10 when T_com was raised from 125 to 195 °C. The onset of porous structure formation took place at close T_com (155 and 160 °C) for snacks made without and with bran inclusion, respectively. At this onset temperature, bran inclusion surged NC by a factor of 6. Wheat bran can be considered as a filler that can enhance nucleation and hence expansion [16]. An increase in melt temperature intensifies bubble nucleation and bubble growth and thus promotes expansion and porous structure (high NC) [14]. The cell density also negatively correlated to predicted melt viscosity (η_com) (*R^2^* > 0.95, Figure 8d), with distinct power function expressed to each bran content. This trend can mainly be attributed to the negative effect of shear viscosity on the expansion mechanisms such as nucleation, bubble growth, and setting [14]. Higher porosity can make a snack texture more fragile/crispy. Therefore, the design of wheat-based snacks requires a compromise between hydrosolubility of starch and porosity of expanded structure.

As an example, we want to manufacture wheat snacks having the following characteristics: (1) moderately expanded (SEI > 4.5), (2) moderately porous, i.e., medium level of cell density (1000 < NC < 2000), and (3) moderate hydrosolubility of starch (50% < WSI_starch_ < 60%). From the correlations between extrusion variables (T_com, η_com) and structure (Figure 8b–d) and the relationship between SME_com and WSI_starch_ (Figure 8a), this kind of snack can be obtained by extruding LB-based recipe, containing 12.6% bran, at T_com of 165–170 °C, η_com < 100 Pa.s, and SME_com > 375 kJ/kg. By examining the feasible region in extruder operating charts (Figure 8e–g), for flow rate Q = 20 kg/h, the targeted T_com, η_com, and SME_com can be achieved by tuning the parameters as follow: a single point of N (700 rpm) for low Tb (125 °C) and small windows of N (630–700 rpm) for higher Tb (140 °C).

In general observation, the 1D global extrusion model is more accurate in the prediction of extruded legumes features than extruded starch ones. The protein structural changes of extruded legumes were predicted from the computed temperature at die exit (Figure 7a,b), while the starch structural changes from computed SME (Figure 5a, Figure 6a,b and Figure 8a). The overestimation of temperature prediction by the extrusion model is much lower than the underestimation of SME prediction (5–10% vs. 35–65%) (Figure 1) and, therefore, a better prediction of extruded legumes features can be expected. The extrusion model neglects the solid transport and solid friction energy and material sticking, leading to SME underestimation. The friction and sticking phenomena depend on the raw materials. It explains why the prediction of intrinsic viscosity of extruded potato starch, through computed SME, is poorer than that of other starches (Figure 5a). Better SME prediction, and thus better product features prediction, could be achieved by improving the extrusion model by taking into account the friction and sticking coefficient.

## 4. Conclusions

Twin-screw extrusion processing of starchy products was simulated using the 1D global extrusion model (Ludovic^®^ software v7.0.0 Classic Edition (Sciences Computers Consultants, Saint Etienne, France)), and melt rheological model as input. A large data set has been built, including results for extruded products’ properties and extruder variables (temperature and specific mechanical energy), taken as output variables. Satisfactory correlations between process parameters predicted extrusion variables and features of extruded foods, such as structural and functional ones, justify the use of the extrusion model as a computer-aided tool for designing starchy food by extrusion. Moreover, the capacity for predicting these features is extremely vital to reducing time and labor costs in industrial foods R&D.

Flow variables at the die exit, such as melt temperature, specific mechanical energy and melt viscosity, can be computed and used, in turn, as the input variables for a phenomenological model of expansion, allowing predicting expansion indices and cellular structure of expanded starchy foods. The possibility to compute melt viscosity is particularly interesting since it is difficult to measure, either in-line or off-line and because it governs the expansion phenomenon, at least for low hydrated starchy foods. In addition, the knowledge of temperature profile and processing conditions resulting from the simulation is useful in setting extruder configuration and operating parameters for extrusion of thermo-sensitive foods.

As a future prospect, extrusion simulation will be applied to the design of a wide range of starch-protein blends and expanded snacks from pulse crops. Further possible extension includes the processing of plant proteins for meat analogues. Besides classical structural and functional properties, the textural and nutritional features, such as starch and protein digestibility, inactivation of anti-nutritional factors, and destruction of pathogenic microbes and vitamins, will be tackled. The rheology database will be updated with melt viscosity models encompassing a wide range of moisture content, temperatures, and SME, taking into account protein and fiber content, as well as protein structural changes and starch-protein morphology.

## Figures and Tables

**Figure 1 foods-11-01780-f001:**
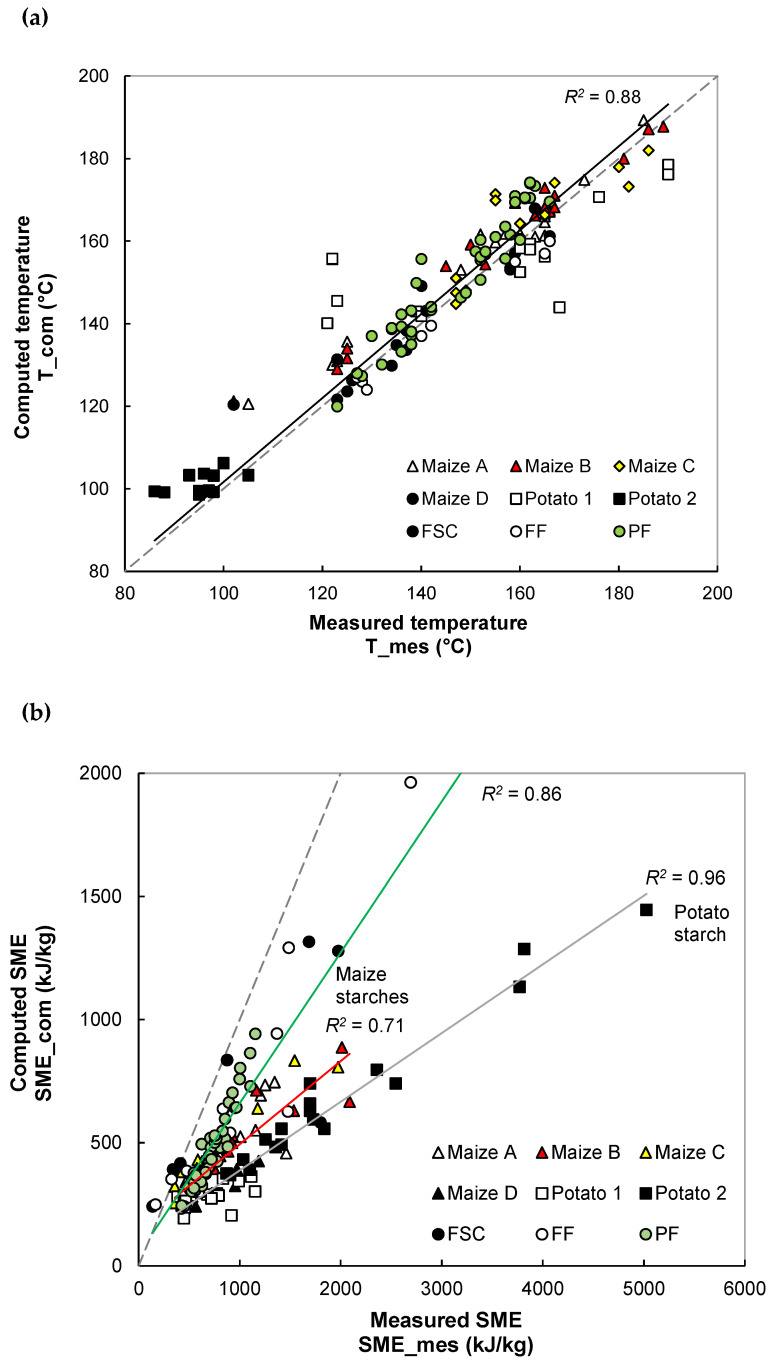
Comparison between predicted and measured extrusion variables: temperature (**a**) and specific mechanical energy (SME) (**b**). The solid lines following linear equations stand for correlation between measured and predicted variables, while the dotted line stands for the equity between them.

**Figure 2 foods-11-01780-f002:**
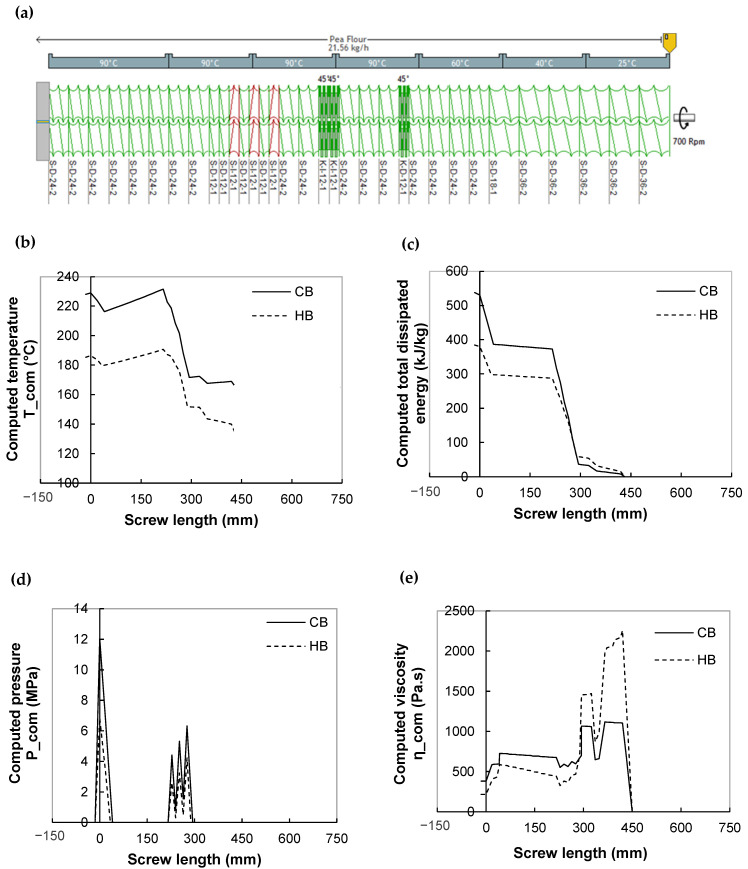
Profiles of screw (**a**) and computed flow variables along the screws for breakfast cereals (CB -) and (HB,--): (**b**) product temperature, (**c**) total dissipated energy, (**d**) pressure, and (**e**) viscosity. Processing conditions: flow rate Q = 20 kg/h, screw speed N = 500 rpm, moisture content MC = 15% and barrel temperature profile: 20, 40, 60, 90, 90, 90, 90 °C.

**Figure 3 foods-11-01780-f003:**
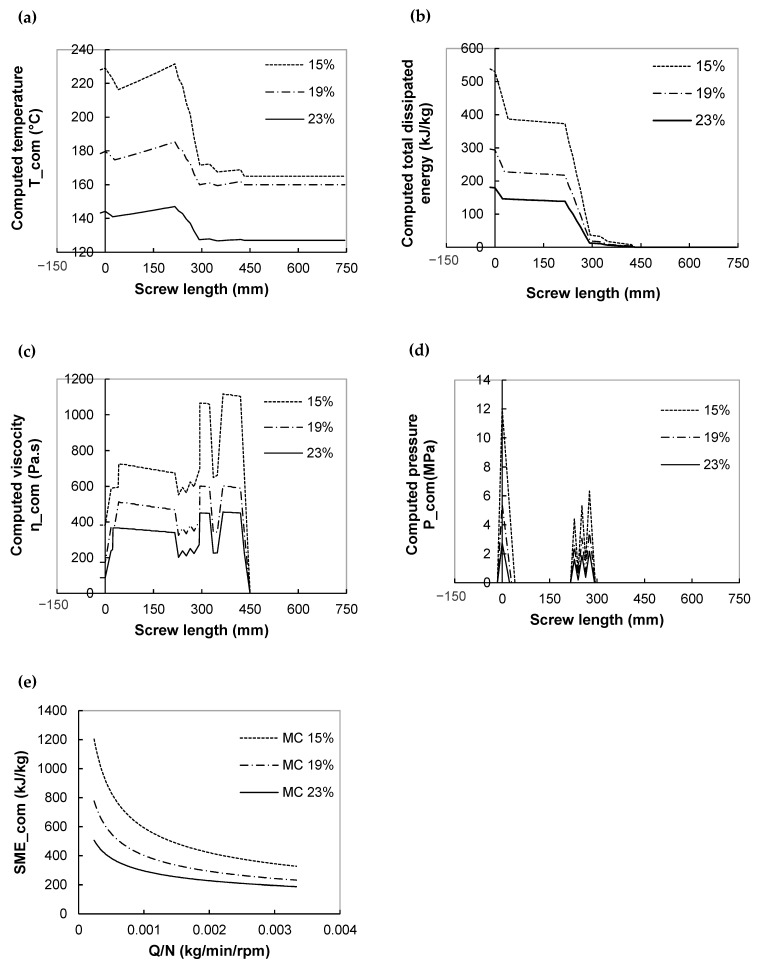
Effect of moisture content on computed temperature (**a**), total dissipated energy (**b**), viscosity (**c**), and pressure (**d**) along the screw and variation of SME with filling ratio (Q/N) (**e**). Study case: breakfast cereal: CB formulation. For (**a**–**d**), processing conditions: Q = 20 kg/h, N = 500 rpm, Tb = 90 °C.

**Figure 4 foods-11-01780-f004:**
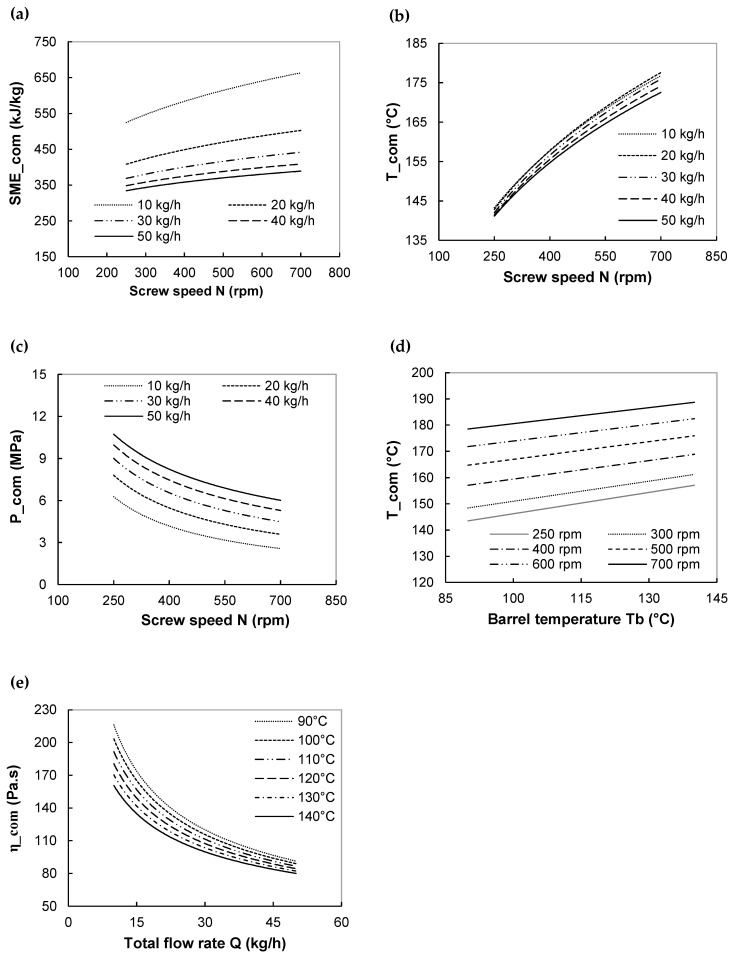
Sensitivity study of the effect of extrusion operating conditions on predicted flow variables for breakfast cereals. HB recipe, MC = 19%. (**a**) variation of SME_com, (**b**) melt temperature at die exit T_com and (**c**) die entrance pressure P_com with screw speed N for various flow rates Q. Last barrel temperature Tb = 90 °C, (**d**) variation of T_com with Tb for various screw speeds N. Q = 20 kg/h, (**e**) variation of viscosity η_com with Q for various Tb. N = 500 rpm.

**Figure 5 foods-11-01780-f005:**
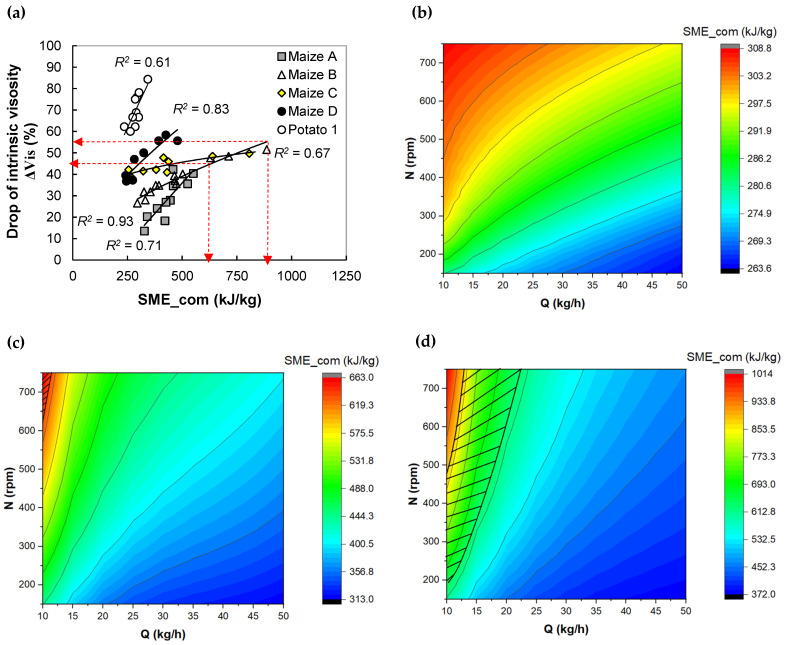
Application of extrusion simulation in the design of expanded maize starches having targeted starch transformation. Step 1. Relationship between the drop of intrinsic viscosity and computed SME (SME_com) (**a**). The arrow indicates the lower and upper limits of targeted SME_com and ΔVis. Step 2. Creation of extruder operating chart by 1D extrusion model. The effect of length of the left-handed element was studied on the SME_com level: (**b**) 0 mm, (**c**) 25 mm, and (**d**) 50 mm. Step 3. Determination of range of extrusion conditions (Q, N), delimited by hatched area, leading to SME_com between 625 and 875 kJ/kg according to the goal: drop of intrinsic viscosity = 50 ± 5% for feed MC = 20% and amylose content = 47%.

**Figure 6 foods-11-01780-f006:**
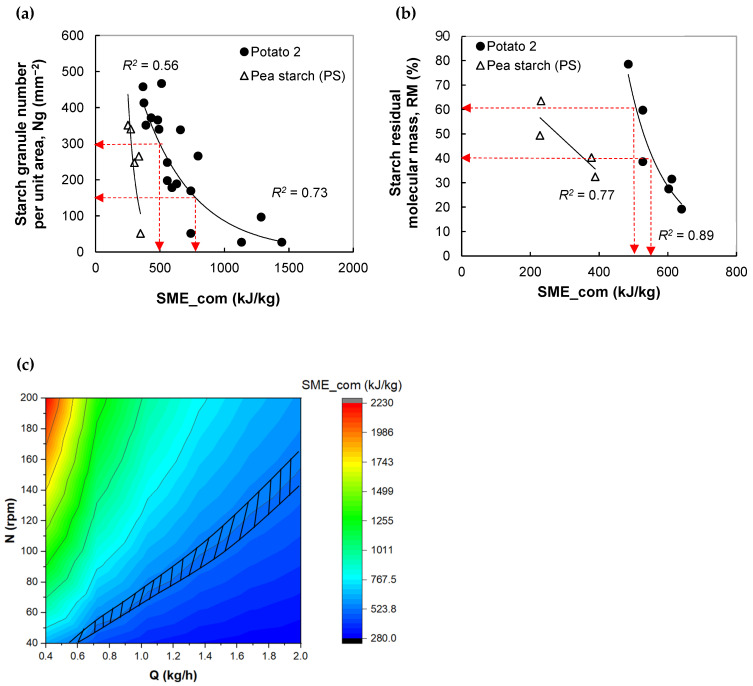
Application of extrusion simulation in the design of dense extruded starches having targeted starch transformation. Step 1. Variation of the remnant granule number per unit area (Ng) (**a**) and residual molecular mass (RM) (**b**) with SME_com. The arrows indicate lower and upper limits of targeted SME_com, Ng, and RM. Step 2. Creation of extruder operating chart by the 1D extrusion model and determination of the range of extrusion conditions (Q, N), delimited with a hatched area, leading to SME_com between 500 and 550 kJ/kg according to the goal: RM = 50 ± 10% and 150 ≤ Ng ≤ 300 (**c**). Fixed conditions: MC = 30%, last barrel temperature = die temperature = 95 °C.

**Figure 7 foods-11-01780-f007:**
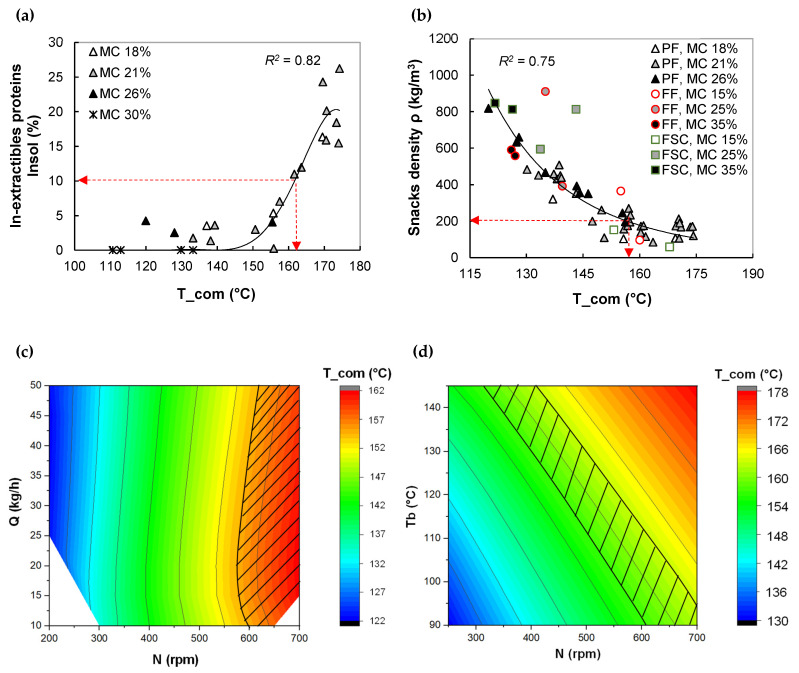
Application of extrusion simulation in the design of pea snacks. Step 1. Variation of in-extractible proteins (**a**) and density (**b**) of snacks with T_com. The arrows indicate the upper limits of targeted T_com and snack properties. Step 2. Creation of extruder operating chart by the 1D extrusion model and determination of the range of extrusion conditions (Q, N, Tb), delimited with a hatched area, leading to predicted melt temperature between 155 and 165 °C according to the goals: snack density <200 kg/m^3^, and protein structural change: insoluble proteins <10% for feed moisture contents of 21% (**c**,**d**).

**Figure 8 foods-11-01780-f008:**
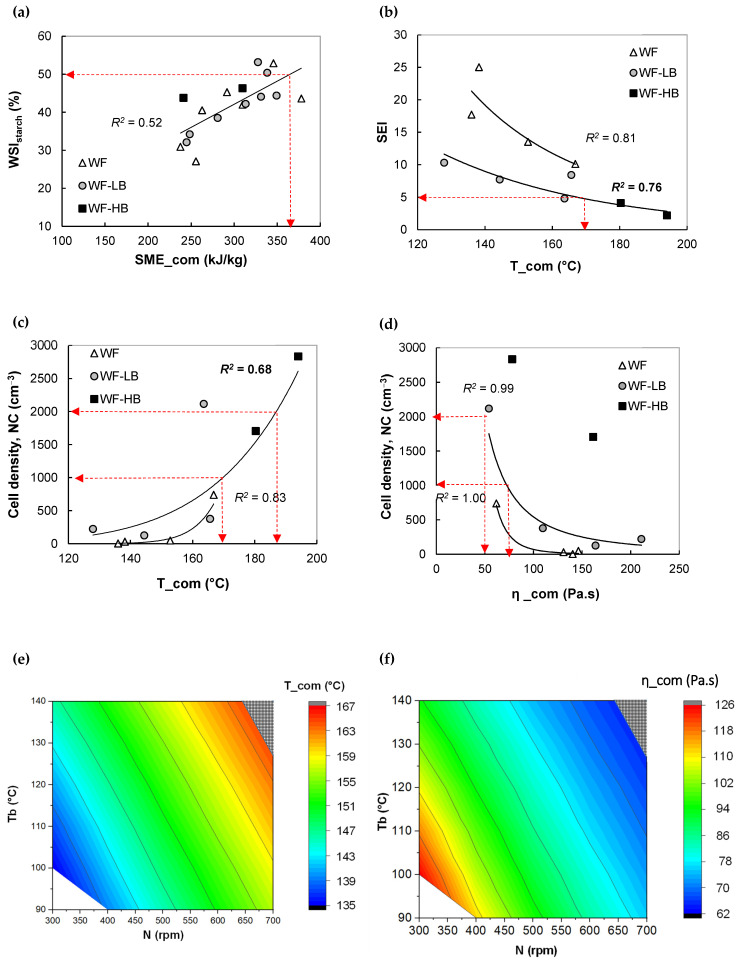
Application of extrusion simulation in the design of wheat snacks. Feed: Wheat flour enriched with low bran content. Step 1. Variation of water solubility index of starch (WSIstarch) with SME_com (**a**), and variation of sectional expansion index (SEI) (**b**) and cell density (NC) (**c**,**d**) with T_com and η_com. In (**a**), the arrows indicate the lower limit of targets, while in (**c**), they indicate lower and upper bounds. Step 2. Creation of extruder operating chart and determination of extrusion parameters range. Fixed operating conditions: MC = 22% and Q = 20 kg/h. The hatched area in (**e**–**g**) indicate the range of extrusion conditions (N, Tb) leading to T_com and SME_com according to the goals: a porous and expanded structure (1000 ≤ NC ≤ 2000 and SEI ≥ 4.5); and 50% ≤ WSI_starch_ ≤ 60%.

**Table 1 foods-11-01780-t001:** Chemical composition of raw material, mostly provided by the supplier.

No	Formulation	Supplier	Chemical Composition							Starch Components	
			Starch	Protein	Fiber	Lipid	Ash	Sugar	Salt	Amylose	Amylopectin
	Expanded starches										
1	Blends of high amylose Eurylon 7 (Maize A) and waxy maize Waxilys 100 (Maize D) starches [27,33]										
	Maize A	Roquette (France)	93.4%	0.15%	<1%	1.3%	0.15%	n.g.	<0.01%	70%	30%
	Maize B: A/D = 2/1		93.0%	0.18%	<1%	1.1%	0.13%	n.g.	<0.01%	47%	53%
	Maize C: A/D = 1/1		92.9%	0.20%	<1%	0.67%	0.13%	n.g.	<0.01%	23.5%	76.5%
	Maize D	Roquette (France)	92.3%	0.25%	<1%	0.20%	0.10%	n.g.	<0.01%	1%	99%
2	Potato starch: Potato 1 [31]	Roquette (France)	95.7%	0.08%	<1%	0.06%	0.60%	n.g.	<0.01%	21%	79%
	Dense starch										
3	Potato starch (Potato 2) [32]	Roquette (France)	95.7%	0.08%	<1%	0.06%	0.6%	n.g.	<0.01%	21%	79%
	Pea starch [32]	Roquette (France)	97.7%	0.49%	<1%	0.79%	0.5%	n.g	<0.01%	35%	65%
4	Breakfast cereals (this work; confidential)										
	Malt extract-based (HB)		82.3%	4.6%	0.52%	1.4%	0.17%	10.0%	1.0%	20–28%	72–80%
	Cacao based (CB)		72.7%	6.3%	1.3%	2.8%	0.8%	10.0%	1.0%	20–28%	72–80%
	Expanded snacks										
	a) Leguminous starch-protein blends										
5	Pea flour (PF) [34]	Sotexpro (France)	46.3%	23.9%	25.7%	2.0%	2.1%	n.g.	n.g.	35.0%	65.0%
6	Fava bean ingredients (this work)										
	Fava bean flour (FF)	Valorex (France)	45.3%	32.8%	6.3%	4.0%	3.7%	3.0%	n.g.	33.5%	66.5%
	Fava bean starch concentrate (FSC)	Valorex (France)	59.2%	20.6%	7.3%	1.7%	2.7%	2.8%	n.g.	39.9%	60.1%
7	Cereal based: Wheat flour type 550 enriched with wheat bran [35,36]										
	WF: Wheat flour	Provimi Kliba S.A. (Switzerland)	78.5%	13.0%	2.8%	1.1%	0.8%	n.g.	n.g.	n.g.	n.g.
	WF-LB: WF + low bran content	Provimi Kliba S.A. (Switzerland)	69.7%	13.9%	12.6%	1.6%	1.8%	n.g.	n.g.	n.g.	n.g.
	WF-HB: WF + high bran content	Provimi Kliba S.A. (Switzerland)	55.5%	15.0%	24.4%	2.2%	3.2%	n.g.	n.g.	n.g.	n.g.

**Table 2 foods-11-01780-t002:** Database of extrusion operating conditions and variables.

	**Formulation Number and Products Code**			
**Specifications**	**1**	**2**	**3**	**4**
	Expanded starch		Dense extruded starch	Breakfast cereal
	Maize A	Potato 1	Potato 2	CB
	Maize B		PS	HB
	Maize C			
	Maize D			
References	[27,33]	[31]	[32]	This work
Extruder brand	Clextral	Clextral	Thermo-scientific	Coperion Werner & Pfleiderer
	BC 45	BC 45	HAAKE™ Polysoft OS	ZSK 26Mc
Screw diameter D (mm)	55.5	55.5	16	25.5
Screw length (L)	1000	1000	632	740
Screw profile	conveying elements	conveying elements	conveying elements	conveying elements
from hopper to die	reverse screw element	reverse screw element	90° bilobal kneading discs	three blocks of 45° kneading discs
			reverse screw element	three reverse screw elements
			conveying element	conveying elements
Die configuration	a twin-slit	two circular dies	conical die	circular die
	rheometric die			
Moisture content MC (w.b.)	20 < MC < 36%	22 < MC < 35%	25 and 30%	15; 19; 23%
Melt temperature °C	135 < T < 190	120 < T < 190	85 < T < 105	confidential
SME (kJ/kg)	250 < SME < 2000	350 < SME < 1250	850 < SME < 5500	confidential
	**Formulation Number and Products Code**			
**Specifications**	**5**		**6**	**7**
	Expanded snacks			
	PF		FF	WF
			FSC	WF-LB
				WF-HB
References	[34]		This work	[35,36]
Extruder brand	Coperion Werner & Pfleiderer		Thermo Scientific™	Clextral
	ZSK 26Mc		Process 11	Evolum 25
Screw diameter D (mm)	25.5		11	25
Screw length (L)	740		269.5	400
Screw profile	conveying elements		conveying elements	conveying elements
from hopper to die	three blocks of 45° kneading discs		90° bilobal kneading discs	mixing elements
	three reverse screw elements		reverse screw element	reverse screw element
	conveying elements		conveying element	
Die configuration	(a) multiple slit die + circular die		circular die	circular die
	(b) circular die			
Moisture content MC (w.b.)	18% ≤ MC ≤ 26%		15; 25; 35%	18 and 22%
Melt temperature °C	120 < T < 165		120 < T < 165	n.g.
SME (kJ/kg)	450 < SME < 1200		135 < SME < 3500	n.g.

**Table 3 foods-11-01780-t003:** Parameters of viscosity model for various molten starchy products.

Coefficients	Maize Starch with Amylose Content of				Potato	Breakfast Cereals		Pea Flour	Wheat Flour + Wheat Bran *		
	70%	47%	23.50%	1%	Starch						
Product Code	Maize A	Maize B	Maize C	Maize D	Potato 1	CB	HB	PF	WF	LB	HB
References	[27]				[31]	[50]		[49]	[37]		
$K_{0}$ (Pa·s^n^)	1.21 × 10^7^	1.91 × 10^6^	4.57 × 10^7^	4.72 × 10^6^	3.67 × 10^6^	1.71 × 10^6^	1.22 × 10^7^	8.1 × 10^5^	8.10 × 10^−2^	1.60 × 10^−1^	2.00 × 10^−3^
$\frac{E}{R}$ (K)	11,440	7983	10,850	9235	6421	6765	11,153	4210	7343	7553	11,201
α	14.6	17.9	29.5	26.1	14.9	13.6	13.7	11.4	23.5	27.4	34.5
β (kg kJ^−1^)	-	7.44 × 10^−5^	1.06 × 10^−3^	1.65 × 10^−3^	3.08 × 10^−3^	-	-	-	1.67 × 10^−3^	1.94 × 10^−3^	7.50 × 10^−3^
$n_{0}$	−1.16	1.81	3.54	−1.02	1 × 10^−2^	-	-	0.29	0.166	0.165	0.129
$\alpha_{1}$ (°C)^−1^	7.93 × 10^−3^	−7.36 × 10^−3^	−1.54 × 10^−2^	7.20 × 10^−3^	1.55 × 10^−3^	3.28 × 10^−3^	3.70 × 10^−3^	-	-	-	-
$\alpha_{2}$	1.31	−1.93	−3.19	2.54	9.96 × 10^−2^	−2.18	−2.55	-	-	-	-
$\alpha_{3}$ (kg·kJ^−1^)	-	−4.89 × 10^−3^	−8.72 × 10^−3^	1.56 × 10^−4^	2.75 × 10^−4^	-	-	-	-	-	-
$\alpha_{4}$ (°C)^−1^	-	-	-	-	1.53 × 10^−1^	8.05 × 10^−3^	9.5 × 10^−3^	-	-	-	-
$\alpha_{5}$ (kg·°C^−1^·kJ^−1^)	-	1.98 × 10^−5^	4.10 × 10^−5^	-	−3.47 × 10^−5^	-	-	-	-	-	-
$\alpha_{6}$ (kg·kJ^−1^)	-	9.19 × 10^−3^	1.12 × 10^−2^	-	−1.27 × 10^−7^	-	-	-	-	-	-

* The authors computed model parameters without taking into account reference variables ($T_{0}$, ${\mathrm{MC}}_{0}$). Therefore, the viscosity model took the form as; $K=K_{o}\exp\left[\frac{E}{R}\left(\frac{1}{T}\right)-\alpha \;\left(\mathrm{MC}\right)-\beta \;\left(\mathrm{SME}\right)\right]$.

**Table 4 foods-11-01780-t004:** Data base of measured structure and properties of extruded products.

No	Product Code	Product Features	Measurement Method and Reference		Database Source
	Expanded starches	Starch destructuration	Ubbelhode viscometer	[27]	[27]
1	Maize A	Intrinsic viscosity			
	Maize B	Melt rheology			
	Maize C	Shear viscosity	On-line twin-slit rheometer	[51]	[27]
	Maize D		(Rheopac)		
		Melting temperature	DSC	[52]	[45,46] *
2	Potato 1	Intrinsic viscosity	Ubbelhode viscometer	[31]	[31]
	Dense starch	Starch depolymerization			
3	Potato 2	Residual molar	Asymmetrical flow field	[32,53]	[32]
	Pea (PS)	mass	flow fractionation (AF4)		
			coupled to MALLS		
		Starch melting			
		Granule number	Light microscopy and image	[32]	
		per unit area	analysis		
		Melting temperature	DSC	[32]	
4	Breakfast cereals	Melt rheology			
	Malt-based (HB)	Shear viscosity	Off-line preshearing	[54]	[50]
	Cacao-based (CB)		capillary rheometer		
			(Rheoplast^®^)		
		Melting temperature	DSC	[48]	This work
	Expanded snacks				
	Legume-based				
5	Pea flour (PF)	Protein aggregation			
		In-extractible proteins	Protein solubility in	[55]	[34]
			SDS + DTE reagents		
		Structure			
		Density	Bead displacement	[34]	[34]
		Melt rheology			
		Shear viscosity	Rheoplast^®^ rheometer	[54]	[49]
		Melting temperature	DSC	[52]	[34]
6	Fava bean ingredients	Structure			
	FF, FSC	Density	Bead displacement	[34]	This work
		Melt rheology			
		Shear viscosity	Rheoplast^®^ rheometer	[54]	[49]
		Melting temperature	DSC	[32]	This work
	Cereal-based				
7	Wheat flour enriched	Structure			
	with wheat bran	Radial expansion	Measurement of diameter	[35]	[35]
	WF, WF-LB, WF-HB	index (SEI)	of extrudate by Caliper		
		Cellular structure	X-ray microtomography and	[35]	[35]
		(Cell number	3D image analyze		
		per cm^3^)	(Component labeling operation)		
		Functional property			
		Water solubility	Solubilisation and centrifugation	[36]	[36]
		index of starch			
		Melt rheology			
		Shear viscosity	On-line twin-slit	[37]	[37]
			rheometer (TSAR)		
		Melting temperature	DSC	[52]	[42] **

* Melting temperature for maize starches having various amylose content was approached by that of other starches found in the literature: waxy maize starch (amylose content of 1%), wheat starch (25%), smooth pea starch (35%), and wrinkled pea starch (75%). ** Melting temperature of bran enriched wheat flour was approached by that of wheat flour found in the literature.

## Data Availability

The data presented in this study are available in Appendix A, and the dataset can also be extracted from the Results Section (Figure 1, Figure 2, Figure 3, Figure 4, Figure 5, Figure 6, Figure 7 and Figure 8, Table 1, Table 2, Table 3 and Table 4).

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
