# Peer review of "Extrusion Simulation for the Design of Cereal and Legume Foods"

_foods, 2022, doi:10.3390/foods11121780_

Round 1

Reviewer 1 Report

lines 173-176: The introduction must be in function of the objective.  So, “introduction” should be about which others research have used 1D model and specifically for extrusion and which were the results.  Add paragraph about those works.

 Line 192-193. Add in text, references of the databases.

 Line 388.  How do you define the melting temperature for a biopolymer?

 How was the methodology to measure the melting temperature? Add in materilas and methods.   

How I know the melting temperature from figure 1 (by example for maize). 

Table 4. add the value of the melting temperature in the table 4.

In fig . 5,6, 8, R2 is low  (0.61 ) This means that the model is not accuracy. How author can improved this R2? Discuss in text.

In your simulation, legumes had the better R2 compared with corn starch and wheat starch. Why?. Discuss  in text. 

Reviewer 2 Report

The manuscript entitled  “Extrusion simulation for the design of cereal and legumes foods” describes using relatively simple numerical model to design various starchy products with targeted structure and properties. It has been proven that by using this model it is possible to design snack products quickly and cheaply. The research was carried out for a group of seven  different starchy food formulations.

The methodology is described correctly. The results are clear and well explained, hovewer there are several tables in the text with very large amounts of information.

In this form, these tables are unreadable, so you should try to present this information in a more compact way.

Table 2 is as large as 3 pages, full of long descriptions, which significantly increases the size of the table. The table should be more readable and compact.

Table 4 will be more readable in vertical layout, due to narrowing of "Measurement method & reference" and "Database source" columns

Fig 4 a, b, c - values on the horizontal axis should be presented identically in all cases

Based on the high quality of the manuscript I suggest it to be accepted after minor revision.

Reviewer 3 Report

The manuscript is about the extrusion simulation for the design of cereal and legume foods. It is well-structured and the results are of great importance and may be very interesting. 

comments:

"Legumes" should change into "Legume" in the title

Both "cocoa" and "cacao" are used in the manuscript. Please use one of them throughout the manuscript.   
